# Physical Activity and Mental Health After COVID-19: The Role of Levels and Domains of Physical Activity

**DOI:** 10.3390/life15081179

**Published:** 2025-07-24

**Authors:** Miloš Stamenković, Saša Pantelić, Saša Bubanj, Bojan Bjelica, Nikola Aksović, Ovidiu Galeru, Tatiana-Nela Balint, Alina-Mihaela Cristuță, Carmina-Mihaela Gorgan, Tatiana Dobrescu

**Affiliations:** 1Faculty of Sport and Physical Education, University of Niš, 18000 Niš, Serbia; kineziologija92@gmail.com (M.S.); spantelic2002@yahoo.com (S.P.); 2Faculty of Physical Education and Sports, University of East Sarajevo, 71126 Lukavica, Bosnia and Herzegovina; vipbjelica@gmail.com; 3Faculty of Sport and Physical Education, University of Priština-Kosovska Mitrovica, 38218 Leposavić, Serbia; kokir87np@gmail.com; 4Faculty of Movement, Sports, and Health Sciences, “Vasile Alecsandri” University of Bacău, 600115 Bacău, Romania; tbalint@ub.ro (T.-N.B.); cristuta.alina@ub.ro (A.-M.C.); gorgan.carmina@ub.ro (C.-M.G.); tatiana.dobrescu@ub.ro (T.D.)

**Keywords:** physical activity level, domains, depression, stress, anxiety, COVID-19

## Abstract

(1) Background: Physical activity (PA) plays a crucial role in preserving and enhancing mental health, particularly in the aftermath of major health crises such as the COVID-19 pandemic. However, the specific levels and domains of physical activity that have the greatest impact on alleviating symptoms of anxiety, depression, and stress in the post-COVID-19 period remain unclear. The aim of this study was to examine the influence of different levels and domains of PA on mental health parameters, specifically symptoms of anxiety, depression, and stress, in individuals who had recovered from COVID-19. (2) Methods: The study included initial measurements (2–4 weeks post-recovery) and final measurements (14–16 weeks post-recovery). The sample comprised 288 participants aged 20 to 60 years (M = 47.06; SD = 12.41), with 95 men and 193 women. PA was assessed using the long version of the IPAQ questionnaire, while mental health was evaluated using the long version of the DASS scale. (3) Results: Stepwise regression analysis revealed that low- (*p* = 0.010) and moderate-intensity (*p* = 0.022) PA was significantly associated with reductions in anxiety symptoms as well as lower stress levels (low PA: *p* = 0.014; moderate PA: *p* = 0.042). Total PA (*p* < 0.001) and vigorous-intensity PA (*p* = 0.008) emerged as significant predictors of reduced depression levels. Among the domains of PA, home-based activities had a statistically significant impact on all three mental health components: anxiety (*p* = 0.005), depression (*p* = 0.002), and stress (*p* = 0.041). Transport-related PA was significantly associated with anxiety (*p* = 0.011) and stress (*p* = 0.022), but not with depression. (4) Conclusions: The results suggest that a combined model incorporating different levels and domains of PA may represent an effective approach to improving mental health in individuals recovering from COVID-19. Further longitudinal studies are needed to establish more precise causal relationships.

## 1. Introduction

The COVID-19 pandemic brought significant changes to daily life and how people function in their environments. In the efforts to limit virus transmission, nearly the entire population experienced quarantine and restrictive measures [1]. These circumstances had a profound impact on quality of life [2], decreased physical activity (PA) levels [3], and negatively affected mental health [4,5]. A marked increase in symptoms of anxiety, depression, and stress was documented as a direct consequence of isolation and social distancing [6,7]. Research has shown that reduced PA led to an increase in symptoms of anxiety and depression [8].

From a health perspective, PA is crucial for maintaining and improving individual and population well-being. Consequently, PA is regarded as an essential lifestyle choice [9]. Numerous studies have documented the positive effects of PA on mental health [10,11,12,13]. For instance, Hamer and associates [13], in a sample of 19,842 participants of both sexes, found that any form of PA was associated with a lower risk of psychological distress. They also noted that different types of activities, including domestic chores, gardening, walking, and sports, were linked to reduced psychological stress levels. Their findings suggested that engaging in at least 20 min of PA per week is effective for maintaining mental health and well-being.

Cecchini and associates [10] found that moderate PA was sufficient to prevent the worsening of depression symptoms during home confinement. Meanwhile, Borrega-Mouguinho and associates [14] observed that vigorous PA performed at home during lockdowns was effective at reducing anxiety, depression, and stress, while also confirming the benefits of moderate-intensity PA.

In terms of PA as a protective immunological factor, studies have highlighted the link between activity levels and the severity of COVID-19 disease [15,16]. Lee and associates [15], in a large sample of 217,768 Korean adults, found a lower risk of contracting COVID-19, as well as reduced severity and mortality, among those who engaged in aerobic and strength activities compared with inactive individuals. Their study further showed that a range of 500 to 1000 MET-min/week was associated with decreased risk of infection, severe illness, and death. Steenkamp and associates [16] also reported that PA levels reduced the risk of hospitalization, intensive care admission, ventilation, and mortality, with better outcomes for those who were physically active.

PA includes a wide range of movements of varying intensity and context, which makes this phenomenon complex to understand from a scientific research perspective. This partially explains the inconsistency in the literature regarding the specific effects of different levels and types of PA on overall health and well-being. Nevertheless, despite this inconsistency, the literature consistently supports the notion that regular PA can improve both physical and mental health compared with physical inactivity. This view is reinforced by a comprehensive umbrella review [17], which found that various levels of PA, especially low and moderate intensity, significantly reduce the risk of depression and anxiety in the general population. This finding strengthens the recommendation that PA, regardless of intensity, is a crucial protective factor for both mental and physical health.

While many studies have examined the positive effects of PA on mental health during the pandemic, relatively few have explored how specific PA domains (e.g., recreational, transport-related, occupational, or household) and intensity levels (e.g., walking, moderate, vigorous) impact psychological outcomes in post-COVID-19 populations.

This study is grounded in the biopsychosocial model, which posits that psychological health is influenced by a complex interaction of biological, behavioral, and social factors [18].

In this framework, PA can serve both as a behavioral regulator and physiological modulator of stress responses, mood regulation, and general well-being. Understanding the specific contributions of PA domains and intensity levels is essential for developing personalized, context-sensitive strategies to mitigate mental health issues following COVID-19 infection.

Previous work by Stamenković and associates [19] found that higher PA intensity (walking, moderate, vigorous, and total) was associated with reduced anxiety and depression among women post-COVID-19. Interestingly, only household PA among the PA domains showed a statistically significant relationship with psychological outcomes.

More recently, Stamenković and associates [20] extended this investigation in a mixed-gender sample and observed improvements in mental health three months after recovery, particularly among older women. These findings suggest that both the type and context of PA may play critical roles in shaping psychological recovery trajectories after COVID-19.

Therefore, the aim of this study was to examine the associations between different intensity levels (walking, moderate, vigorous, and total PA) and specific PA domains (recreational, occupational, transport-related, and household) with symptoms of anxiety, depression, and stress in individuals who had recovered from COVID-19.

Our main hypothesis posited that various levels and domains of PA would serve as statistically significant predictors of anxiety, depression, and stress symptoms in individuals recovering from COVID-19, with specific levels and domains exerting distinct effects on mental health outcomes.

## 2. Materials and Methods

This study employed a longitudinal design to track changes in PA and mental health across different time points following recovery from COVID-19.

### 2.1. Participants

Prior to conducting the study, we estimated the minimum required sample size using the pwr.t.test() function from the pwr package in R software (v4.4.0). For this calculation, we assumed a medium effect size (Cohen’s d = 0.5), with a significance level of α = 0.05 and a statistical power of 0.80, based on commonly accepted standards for behavioral science research. The calculation indicated that a minimum of 102 participants per group would be needed for a two-tailed independent samples *t*-test. Given that our sample included 193 women and 95 men (total n = 288, Table 1), the total sample exceeded the minimum requirement and provided sufficient power to detect medium-sized effects in group comparisons and regression analyses. Therefore, we consider the sample size statistically adequate for the research aims.

While the initial plan was to include 320 participants, the final sample consisted of 288 participants aged 20 to 60 years (M = 47.06; SD = 12.41), with 95 men and 193 women. The reduction in the sample size resulted from the application of predefined inclusion and exclusion criteria, as detailed in Figure 1, which illustrates the participant selection process.

The study was conducted from February to May 2022. The initial measurement took place after participants had recovered from an infection caused by the Omicron variant of the COVID-19 virus, which was the most contagious strain at the time. The final measurement was carried out three months later.

### 2.2. Inclusion/Exclusion Criteria

To participate in this study, individuals had to meet specific criteria. Clear inclusion and exclusion criteria were established to ensure that participants met the requirements for study enrollment.

Inclusion criteria:Participants of both sexes, aged 20 to 60 years;Confirmed SARS-CoV-2 infection, verified by RT-PCR or antigen testing;Treatment at home without hospitalization, or hospitalization for no longer than seven days, with no need for invasive mechanical ventilation;Initial assessment conducted within 30 days of hospital discharge or the end of self-isolation;No pre-existing mental health diagnoses requiring specific psychiatric pharmacotherapy;Ability to independently complete questionnaires and willingness to participate voluntarily.

Exclusion criteria:Severe clinical presentation of COVID-19;Use of mechanical ventilation (respiratory support) during treatment;Lack of laboratory-confirmed SARS-CoV-2 infection;Individuals not treated at home;Hospitalized patients whose hospital stay exceeded seven days;Participants for whom more than 30 days had passed since hospital discharge or the end of self-isolation at the time of study enrollment;Individuals with diagnosed mental health disorders requiring specific pharmacotherapy.

### 2.3. Ethical Considerations

Before testing began, all participants were informed about the potential benefits and risks of the study and their participation. Informed consent was obtained from each participant, and they were reminded of their right to withdraw at any time.

### 2.4. Methodological Overview of the Study

Data were collected using a telephone interview. Due to epidemiological restrictions during and after the COVID-19 pandemic, in-person data collection was not feasible. Consequently, telephone interviews were employed as a safe and practical alternative. This method was particularly appropriate for participants who lacked stable internet access or sufficient digital literacy—barriers commonly present among older adults and individuals in rural areas within the target population.

Each participant was interviewed twice, with a sufficient interval between the first and second administration of the IPAQ (International Physical Activity Questionnaire) to minimize the potential influence of initial responses on repeated testing results. Participants reported their PA levels during the previous seven days, including the number of days and duration of vigorous and moderate-intensity activities, as well as walking, across all assessed domains [21].

All questionnaires were coded to facilitate linking of baseline and follow-up responses and to ensure accurate comparison of results. Personal data were securely protected and never misused. Participants were fully informed about the research procedures and the need for follow-up testing three months after the initial measurement. The study was conducted in collaboration with the Health Center Leskovac and the General Hospital Leskovac.

Data on participants were collected with the approval of the ethics committees of these healthcare institutions. Participants were selected using stratified random sampling based on their place of residence within the Jablanica District. This approach ensured territorial diversity and representativeness of the sample for southern Serbia.

The study was conducted in accordance with the Declaration of Helsinki and relevant ethical guidelines [22], with approval from the ethics committees of the University of Niš, the Health Center Leskovac, and the General Hospital Leskovac.

### 2.5. Physical Activity Assessment

The PA of participants was assessed using the self-report method with the long version of the International Physical Activity Questionnaire (IPAQ), a widely used and validated instrument for scientific research. This version of the IPAQ has also been psychometrically tested in the Serbian population [21]. Milanović and associates [21], in a study involving 660 older adults, reported moderate to high test–retest reliability (ICC = 0.71–0.74 for total PA; ICC = 0.91 for transport-related PA; ICC = 0.53 for leisure-time PA), confirming its suitability for epidemiological and public health research in Serbia.

Key parameters analyzed included frequency, intensity, and duration of PA. The questionnaire covered eight domains and levels of PA: (a) PA at work; (b) transport-related PA; (c) leisure-time PA; (d) home and garden activities; (e) low-intensity PA; (f) moderate-intensity PA; (g) vigorous-intensity PA; (h) total PA.

The long version of the IPAQ questionnaire enables a comprehensive assessment of PA levels across different contexts, detailing the duration of activities in all eight domains and the number of days per week these activities are performed. This expanded version provides a more methodologically precise evaluation compared with the short version. For each activity, a metabolic equivalent (MET) value was calculated to accurately quantify and categorize PA levels. The total MET-min/week value was obtained by summing the MET values across all domains.

Results were expressed in MET-min/week. To calculate these numerical values, participants reported the total number of minutes per day they engaged in PA and the number of days per week. These values were then multiplied by MET coefficients corresponding to the activity’s intensity. MET coefficients were calculated for each type of activity. For example, different forms of walking were analyzed to determine an average MET value, and the same approach was applied to moderate- and vigorous-intensity activities (Table 2).

The MET coefficients and formulas used in Table 2 were adapted from the official scoring protocol of the IPAQ Research Committee [23], which standardizes the estimation of energy expenditure in MET-minutes per week across different physical activity intensities.

### 2.6. Depression, Anxiety, and Stress Scale

The Depression, Anxiety, and Stress Scale (DASS) was used to assess negative emotional states [24]. The DASS contains 42 items rated on a 4-point scale designed to better define, understand, and measure prevalent and clinically significant emotional states such as depression, anxiety, and stress. Each of the three DASS subscales contains 14 items, further divided into subscales of 2–5 items each, grouped by related content:

The Depression Scale assesses dysphoria, hopelessness, devaluation of life, self-deprecation, lack of interest/involvement, anhedonia, and inertia.

The Anxiety Scale assesses autonomic arousal, skeletal muscle effects, situational anxiety, and subjective experience of anxious affect.

The Stress Scale is sensitive to chronic, non-specific arousal levels. It evaluates difficulty relaxing, nervous arousal, being easily upset/agitated, irritability/over-reactivity, and impatience.

Participants rated the intensity and frequency with which they experienced each emotional state in the previous week on a four-point scale. Total scores for depression, anxiety, and stress were obtained by summing the scores for the relevant items.

The DASS has demonstrated excellent psychometric properties [24]. The Serbian version of the DASS has been psychometrically evaluated by Jovanović and associates [25] using the DASS-21. Their study confirmed a robust three-factor structure and excellent internal consistency (depression α = 0.83; anxiety α = 0.82; stress α = 0.87), supporting the validity and reliability of the instrument in Serbian-speaking populations. Although their study focused on the short form, the items are identical to those in the DASS-42, making it appropriate for use in the present study.

### 2.7. Statistical Analysis

Data were analyzed using SPSS statistical software (version 23.0, SPSS Inc., Chicago, IL, USA). Paired-samples Student’s *t*-test was used to compare the means between the initial and final measurements. Since the paired samples Student’s *t*-test is based on the one-sample *t*-test, we considered the sample size to be sufficiently large for the test statistic to approximately follow a distribution close enough to the normal distribution. Therefore, we did not check the normality of the distributions.

To determine the extent of the impact of PA on mental health parameters at the initial and final measurement points, stepwise linear regression analysis was performed. The significance level for all statistical tests was set at *p* < 0.05. As a preliminary step in the stepwise linear regression, potential multicollinearity among the predictors was examined. A strong correlation was identified only between total PA and moderate PA; therefore, the variable with the weaker explanatory power for the dependent variable was excluded from the analysis.

## 3. Results

The results of the paired samples *t*-test (Table 3) indicate a statistically significant difference between the initial and final measurements for moderate (*p* = 0.001), low (*p* = 0.006), and total PA (*p* < 0.001). No statistically significant difference was found for vigorous PA (*p* = 0.423).

Regarding the domains of PA, the *t*-test showed statistically significant differences only in PA during transportation (*p* = 0.025) and in household PA (*p* = 0.001). No significant differences were found for occupational PA (*p* = 0.372) or leisure-time PA (*p* = 0.477).

In terms of mental health, the *t*-test results revealed a statistically significant difference between the initial and final measurements only for anxiety (*p* = 0.047). There were no statistically significant differences for depression (*p* = 0.266) or stress (*p* = 0.418).

Cohen’s d values in Table 3 indicate small effect sizes for all significant changes, suggesting that while the differences in PA and anxiety levels between the initial and final measurements were statistically significant, the magnitude of these changes was modest in practical terms. The largest effect was observed for total PA (d = 0.236), while changes in mental health variables showed particularly small effects (all d < 0.12).

At the initial measurement, only low-intensity PA showed a statistically significant impact on all three components of mental health: anxiety (*p* = 0.008), depression (*p* < 0.001), and stress (*p* = 0.003). Higher engagement in low-intensity PA was associated with lower symptom levels (Table 4).

This type of modeling, which assumes a linear association, typically yields a low level of explained variance in the dependent variable. Nevertheless, such models still demonstrate statistically significant predictive power. Importantly, the exploration of nonlinear and more complex associations is currently being pursued as part of the authors’ ongoing research agenda.

Concerning Table 5, model 4, which includes low and moderate PA, statistically significantly explains variance in anxiety levels (R^2^ = 0.037). Both predictors had a negative and statistically significant impact, indicating that greater involvement in these activities is associated with reduced anxiety.

In Model 5, the inclusion of vigorous PA alongside total PA increased the explained variance in depression (R^2^ = 0.057). Total PA had a negative impact on depression (β = −0.282), while vigorous PA showed a positive beta value (β = 0.182), suggesting a more complex relationship that may include overexertion effects.

Model 6, incorporating low and moderate PA as independent variables, explains 3.2% of the total variance in stress levels (R^2^ = 0.032). Both predictors showed a negative and statistically significant influence (*p* < 0.05), underscoring their potential protective role in reducing stress.

At the initial measurement, only transport-related PA showed a statistically significant association with anxiety (*p* = 0.001), depression (*p* = 0.001), and stress (*p* = 0.023). Other domains of PA (work activity, household activity, and leisure activity) did not show significant correlations with any mental health component (*p* > 0.05, Table 6).

At the final measurement, PA in the household and transport domains showed statistically significant effects on mental health parameters, but these effects were not consistent across all components (Table 7).

In the anxiety prediction model (Model 10), both household PA (β = −0.165, *p* = 0.005) and transport activity (β = −0.147, *p* = 0.011) had a negative and statistically significant impact, suggesting that greater involvement in activities like housework and commuting is associated with lower anxiety levels.

For depression (Model 11), only household PA showed a significant influence (β = −0.185, *p* = 0.002).

In the stress prediction model (Model 12), both transport activity (β = −0.134, *p* = 0.022) and household activity (β = −0.120, *p* = 0.041) demonstrated a significant protective effect. Participants who were physically active in these domains reported lower perceived stress levels.

## 4. Discussion

Differences in certain levels and domains of PA, as well as in mental health parameters, were observed between the initial and final measurements in our study. The results support our assumption that during the four-month period, participants increased their activity in some domains and levels of PA, which had a positive effect on certain mental health parameters. This is further supported by the increase in average MET values across all PA domains and intensity levels—moderate, low, vigorous, total, transport, household, occupational, and leisure-time—even though not all changes reached statistical significance. In addition, the effect sizes (Cohen’s d) included in Table 3 provide a better understanding of the magnitude and clinical relevance of observed changes. This finding may indicate gradual behavioral recovery and resumption of daily routines among individuals post-COVID-19. These results suggest the potential clinical relevance of even small, positive changes in physical activity patterns and psychological functioning.

Similar findings were reported by Zhao and associates [26], who, in a longitudinal study of adults in Australia, found that increased PA over several months was associated with significant improvements in mental health parameters, including depression, anxiety, and stress.

On the other hand, no statistically significant differences were found in vigorous PA, which may reflect difficulties in maintaining high-intensity PA over time, particularly in populations recovering from COVID-19. Previous research has shown that individuals with post-COVID syndrome exhibit significantly reduced aerobic capacity and muscle strength, which can limit participation in demanding forms of exercise [27]. Nevertheless, a modest increase in the average MET value for vigorous PA was recorded, suggesting a potential trend toward recovery in this domain (Table 3).

Furthermore, although a statistically significant reduction was observed in anxiety scores (*p* = 0.047), the absolute mean difference was relatively small (mean = 0.94), which raises the issue of clinical relevance. While statistically meaningful, such a minimal change may not represent a substantial improvement in daily psychological functioning. Future studies should consider reporting minimal clinically important differences (MCIDs) to assess real-world effects of PA on mental health.

Finally, although depressive and stress symptoms declined slightly between measurements, these changes were not statistically significant. This finding could indicate that such symptoms are more resistant to short-term interventions and may require longer durations or higher intensities of PA exposure. Meira and associates [28] emphasized that the frequency and duration of activity are key moderators of psychological outcomes.

### 4.1. Impact of Physical Activity Levels and Domains on Anxiety, Depression, and Stress

A comparison of the results revealed that at the final measurement, certain levels and domains of PA, as measured by the IPAQ questionnaire, had a statistically significant impact on mental health, unlike at the initial measurement. Low-intensity PA, such as walking, and moderate-intensity activity were significantly associated with lower levels of anxiety and stress (Model 4), while vigorous and total PA had no statistically significant impact on anxiety. This is consistent with the findings of a meta-analysis by Xu and associates [29], who showed that walking positively affects mental health regardless of format (group or individual), duration, or location. Regarding moderate-intensity PA, our findings align with previous studies highlighting its positive impact on mental health. Research has shown that moderate PA can reduce symptoms of anxiety regardless of gender, age, or health status [30,31,32].

Moderate activities, such as brisk walking or recreational exercise, activate psychophysiological mechanisms involving neurotransmitter regulation and reduced physiological arousal, contributing to better emotional well-being [33]. Our results support the hypothesis of a protective effect of moderate PA on anxiety symptoms, especially in the recovery period following COVID-19.

In terms of depression, the final measurement results indicated that total and vigorous PA were statistically significant predictors (Model 5). However, the positive beta coefficient for vigorous PA (β = 0.182) suggests a possible paradoxical effect, i.e., higher levels of vigorous activity may be associated with increased depressive symptoms. While most studies highlight the protective effect of vigorous activity [14,34,35], some research warns about the negative effects of excessive PA, particularly in vulnerable populations [36] such as those recovering from COVID-19. Shimura and associates [36] concluded that optimal PA in terms of duration, intensity, and frequency contributes to better psychological outcomes. Conversely, inadequately adjusted or prolonged PA may negatively impact mental health and overall well-being.

Our findings suggest that while total PA may exert a protective influence, the relationship between vigorous PA and depression might not be linear. It is possible that overly intense or prolonged physical activity in certain individuals leads to overexertion, emotional fatigue, or physiological stress, thereby increasing the risk of depressive symptoms. This observation aligns with the hypothesis of a U-shaped association between PA intensity and mental health, where both inactivity and excessive intensity could be detrimental. However, this hypothesis was not statistically tested in the current models. Future studies should incorporate interaction or quadratic terms to examine nonlinear associations and better capture the complexity of the PA–mental health relationship in post-COVID-19 populations.

Overall, the data imply that individuals may respond differently to PA after COVID-19, depending on personal, psychosocial, or physiological factors, highlighting the need for more tailored PA recommendations in post-COVID-19 recovery strategies. These findings align with a growing body of evidence highlighting the complexity of post-COVID-19 recovery and the necessity of individualized approaches. For instance, Mitroi et al. [37] demonstrated that the combination of prior COVID-19 infection and sociodemographic characteristics significantly influenced quality of life among tuberculosis patients, a population already burdened by chronic illness and social vulnerability. Their study emphasizes the relevance of contextual factors, such as disease history, economic status, and social support, when assessing recovery outcomes. Similarly, Cioboata et al. [38] found that both the clinical form of initial COVID-19 illness and the presence of comorbidities were strongly associated with the persistence and severity of post-COVID syndrome symptoms. Although the comorbidities were not statistically analyzed as independent predictors, the authors acknowledged their potential relevance. These findings suggest that the effectiveness of physical activity may be mediated or even constrained by pre-existing health conditions and the overall clinical profile of the individual. Accordingly, our results, although obtained from a community-based sample, should be interpreted with caution and viewed as part of a broader clinical landscape in which PA interventions must be tailored to the individual’s physiological, psychological, and social circumstances.

Regarding stress, our results indicate that low-intensity PA in the form of walking and moderate activity were statistically significantly associated with lower stress levels (Model 6), consistent with previous research [11,39]. Although several PA domains and intensity levels emerged as statistically significant predictors of mental health outcomes, the explained variance in all regression models was relatively low (R^2^ = 1.8–5.7%), indicating that while PA contributes to psychological well-being, many other unmeasured factors, such as personality traits, social support, and sleep quality, likely play a larger role. Future research should include a wider range of variables to enhance explanatory power and model robustness.

The transport domain of PA showed a statistically significant impact on all three mental health components (anxiety, depression, and stress) at the initial measurement (Table 6). At the final measurement (Table 7), home- and transport-related PA predicted anxiety (Model 10), home activity alone predicted depression (Model 11), and both categories predicted stress (Model 12). These results suggest that transport-related PA, as a daily and accessible form of movement, may positively affect mental health, which has been confirmed by earlier studies [40].

However, in our study, the impact of transport-related PA was not uniform for all psychological outcomes, as its effect on stress was not statistically significant. This suggests that different forms of transport-related activity (e.g., walking, cycling, using public transportation with walking) may not equally affect all aspects of mental health. Further research is needed to explore the specific mechanisms through which transport-related PA influences anxiety, depression, and stress symptoms in individuals recovering from COVID-19.

Regarding household PA, research suggests that tasks like cleaning, cooking, and home maintenance can positively impact mental health. For example, a study by Koblinsky and associates [41] found that higher household activity levels were associated with greater gray matter volume in the brain among older adults, potentially indicating better cognitive function. Additional studies have confirmed the link between household PA and reduced stress levels [13].

The explanation for why household activity affects all three mental health parameters may lie in its variety, frequency, and integration into daily life. Hammar and associates (13) support the idea that diverse activities, such as household chores and gardening, are associated with better mental health, provided they last at least 20 min. Additionally, the positive influence of household- and transport-related PA observed in our models may be explained by factors such as daily structure, perceived autonomy, and a sense of purpose. Unlike formal exercise, these domains often reflect meaningful and routine-based engagement, which may support emotional regulation and psychological recovery.

In contrast, the absence of an impact from workplace PA on mental health parameters may be due to the fact that workplace activity is often not a matter of personal initiative, but rather an obligation or burden, diminishing its psychological benefit [40].

In terms of leisure-time PA, the same authors [40] confirmed its positive impact on mental health, though our study did not support this, likely due to irregular or limited participation, which may have reduced its effect.

In general, our study’s findings confirm that levels and domains of PA do not always show consistent and stable impacts on mental health and well-being, especially in individuals who have recovered from COVID-19. Variations observed between the two measurement points suggest that each domain and level of PA may differently influence symptoms of anxiety, depression, and stress depending on the period and the participants’ life context. This underscores the importance of studying PA in a dynamic timeframe for a more accurate understanding of its effects.

When interpreting the results, it is essential to consider the potential influence of personal perception and motivation on different levels and domains of PA. Individual perception of activity, as well as intrinsic motivation, can play a crucial role in how PA affects psychological well-being. For example, research among older adults found that internal motivation for PA was significantly associated with life satisfaction and regular exercise [42]. Satisfying basic psychological needs such as autonomy, competence, and connectedness contributes to better mental health and a higher likelihood of continued PA. Additionally, a study among healthy middle-aged adults found that perceptions of one’s own PA are often inaccurate [43].

Many participants overestimated or underestimated their activity levels, which affected their mental health. Accurate perception of PA was linked to better psychological outcomes and greater life satisfaction.

Taking all of this into account, it is important to emphasize the need for a personalized approach to promoting PA for mental health improvement. Recommendations that consider individual motivational factors and perceptions of activity may be more effective in encouraging long-term engagement in PA and enhancing psychological well-being.

### 4.2. Strengths of the Study

This study has several important strengths. The dual measurements (initial and final) allowed for the monitoring of changes in PA and mental health over time, increasing the validity of the findings. The focus on a population that has recovered from COVID-19 makes the study highly relevant in the context of contemporary public health challenges. Additionally, analyzing multiple levels and domains of PA (home, transport, work, leisure) provided a differentiated understanding of how these parameters affect psychological outcomes.

### 4.3. Limitations of the Study

Despite its strengths, this study has several important limitations that should be considered when interpreting the results. Using a self-report PA questionnaire (IPAQ) may introduce response bias, and the absence of a control group prevents direct comparisons with individuals who did not experience COVID-19. This limits our ability to isolate the effects of COVID-19 recovery on mental health outcomes. Including appropriate comparison groups in future studies, such as individuals with no history of COVID-19 or those recovering from other illnesses, would allow for more robust causal inference. Such designs would help determine whether observed psychological changes are specifically attributable to COVID-19 recovery or reflect broader population-level trends. Longitudinal studies with matched controls could further strengthen the validity and generalizability of future findings.

Furthermore, other psychosocial factors that could affect mental health, such as social support and living conditions, were not considered in this study. The sample included only participants from the Jablanica District, limiting the generalizability of the results to other regions. Moreover, individuals who were hospitalized with severe COVID-19 were excluded from the study, further narrowing the generalizability to those with milder or moderate disease courses. Additionally, the sample was predominantly female (193 women vs. 95 men), which may introduce gender bias and limit the applicability of the findings to broader populations.

In addition, although the DASS-42 is a validated instrument, it measures emotional states over the previous seven days. Therefore, administering it at only two time points may not fully capture the psychological fluctuations that occur throughout the recovery process. Finally, while self-report instruments were practical and widely used during the pandemic, the inclusion of objective data (e.g., wearable devices or physiological markers) would further strengthen the methodological rigor and accuracy of the findings.

Furthermore, although we used change scores to assess improvements in mental health outcomes, we did not adjust for baseline values as covariates in the regression models, which may introduce statistical artifacts such as regression to the mean. Future studies could improve the methodological robustness by including baseline scores as control variables and exploring nonlinear effects through quadratic modeling techniques.

Finally, several methodological issues warrant further attention. First, multiple statistical tests were conducted across various PA domains and intensity levels without applying corrections for multiple comparisons (e.g., Bonferroni adjustment), which increases the risk of Type I error.

Second, although stepwise linear regression was used to explore predictors of mental health outcomes, this approach is known to carry risks of overfitting and model instability. No cross-validation or robustness checks were performed to confirm the reliability of the derived models. These factors may limit the generalizability and reproducibility of our findings. Future research should adopt more conservative and theory-driven statistical methods and aim to replicate findings using alternative modeling strategies and validation techniques.

### 4.4. Future Directions

In light of the current findings and the limitations acknowledged, future studies should aim to include adequate control groups in order to better determine the specific impact of COVID-19 recovery on physical activity patterns and mental health outcomes. The implementation of objective measurement methods, such as accelerometers or physiological tracking tools, would improve data accuracy and reliability. It is also important to include more diverse populations, taking into account age, sex, comorbid conditions, and socioeconomic background, to enhance the generalizability of the results. Finally, studies with longer follow-up periods are needed to capture long-term trends and provide deeper insight into how physical activity may contribute to post-COVID-19 recovery and overall quality of life.

## 5. Conclusions

The results of our study suggest that certain levels and domains of PA, particularly low- and moderate-intensity activity, may be associated with reduced symptoms of anxiety and stress among individuals who have recovered from COVID-19. Domain-specific forms of PA, such as household- and transport-related activity, also appear to contribute uniquely to psychological well-being.

These findings highlight the importance of promoting accessible, daily-life physical activities in mental health recovery strategies for post-COVID-19 populations.

Regarding depression, total PA showed a significant association, while vigorous PA demonstrated a less consistent relationship. Although this might suggest that the influence of high-intensity activity is not linear, such interpretations should be made with caution, as the current study was not designed to test nonlinear effects.

From a public health standpoint, these results emphasize the value of tailoring PA recommendations to individual capabilities, preferences, and life circumstances. Future studies should adopt longitudinal designs, include objective PA measurements, and explore a wider range of psychological, physiological, and contextual variables to better understand the mechanisms linking PA to mental health outcomes.

Overall, our findings support the inclusion of physical activity, especially low- and moderate-intensity forms embedded in daily routines, as a viable component of mental health recovery programs for individuals recovering from COVID-19.

## Figures and Tables

**Figure 1 life-15-01179-f001:**
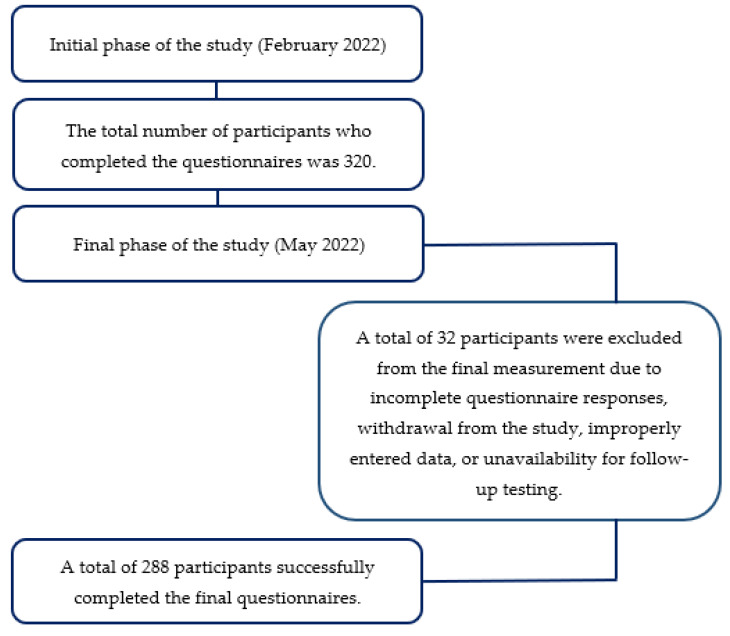
Flowchart of participant inclusion and exclusion.

**Table 1 life-15-01179-t001:** Sociodemographic data of male and female participants aged 20–60 years (n = 288).

**Men (n = 95)**					
Education	Primary	Secondary	University	Master’s/PhD	
	0	38 (40%)	51 (53.7%)	6 (6.3%)	
Residence	Rural	Urban			
	19 (20%)	76 (80%)			
Status	Student	Student	Employed	Unemployed	Retired
	2 (2.1%)	3 (3.2%)	81 (85.3%)	2 (2.1%)	7 (7.4%)
Smoking status	Smoker	Non-smoker			
	25 (26.3%)	70 (73.7%)			
Marital status	Married	Single			
	68 (71.6%)	27 (28.4%)			
**Women (n = 193)**					
Education	Primary	Secondary	University	Master’s/PhD	
	3 (1.6%)	93 (48.2%)	92 (47.7%)	5 (2.6%)	
Residence	Rural	Urban			
	50 (25.9%)	143 (74.1%)			
Status	Student	Student	Employed	Unemployed	Retired
	1 (0.5%)	5 (2.6%)	183 (94.8%)	1 (0.5%)	3 (1.6%)
Smoking status	Smoker	Non-smoker			
	58 (30.1%)	135 (69.9%)			
Marital status	Married	Single			
	147 (76.2%)	46 (23.8%)			

**Table 2 life-15-01179-t002:** Example of MET coefficient calculation.

Type of Physical Activity	Formula for Calculating MET-min/week
Walking (low-intensity activity)	MET-min/week = 3.3 × duration of walking (in minutes) × days walking
Moderate physical activity	MET-min/week = 4.0 × duration of moderate activity (in minutes) × days of moderate activity
Vigorous physical activity	MET-min/week = 8.0 × duration of vigorous activity (in minutes) × days of vigorous activity
Total physical activity	MET-min/week = sum of MET values for walking, moderate, and vigorous activities

Legend: MET—metabolic equivalent, min—minutes.

**Table 3 life-15-01179-t003:** Comparative paired samples *t*-test analysis of mean values for levels and domains of physical activity and mental health parameters between the initial and final measurement, including effect sizes (n = 288).

Pair	Physical Activity Parameters	MET (Initial–Final)	Mean	SD	t	*p*	Cohen’s d
1	Moderate PA (I–F)	2049–2287	−237.77	1248.24	−3.23	0.001	0.190
2	Low PA (I–F)	1007–1121	−113.20	698.05	−2.75	0.006	0.162
3	Vigorous PA (I–F)	823–865	−41.56	878.58	−0.80	0.423	0.047
4	Total PA (I–F)	3869–4273	−403.96	1713.81	−4.00	<0.001	0.236
5	Occupational PA (I–F)	1294–1355	−60.89	1155.81	−0.89	0.372	0.053
6	Transport PA (I–F)	399–461	−62.82	472.09	−2.25	0.025	0.133
7	Household PA (I–F)	1220–1431	−210.33	1019.92	−3.50	0.001	0.206
8	Leisure-time PA (I–F)	968–1005	−36.44	868.68	−0.71	0.477	0.042
	**Mental Health**	**DASS**	**Mean**	**SD**	**t**	** *p* **	**Cohen’s d**
9	Anxiety (I–F)	9.57–8.62	0.94	8.05	1.99	0.047	0.118
10	Depression (I–F)	8.49–7.89	0.60	9.15	1.11	0.266	0.066
11	Stress (I–F)	14.20–13.78	0.42	8.92	0.81	0.418	0.048

Legend: Mean—mean value; PA—physical activity; I—initial measurement; F—final measurement; t—t-statistic; *p*—statistical significance; Cohen’s d—effect size; MET—metabolic equivalent of task for physical activity; SD—standard deviation; DASS—Depression, Anxiety, and Stress Scale.

**Table 4 life-15-01179-t004:** Levels of physical activity as predictors of anxiety, depression, and stress at the initial measurement (n = 288).

**Anxiety—Initial Measurement**							
**Model**	**Predictors**	R	R^2^	F	β	t	*p*
1	Low PA	0.155	0.024	7.026	−0.155	−2.651	0.008
**Depression—Initial Measurement**							
**Model**	**Predictors**	R	R^2^	F	β	t	*p*
2	Low PA	0.212	0.045	13.443	−0.212	−3.666	<0.001
**Stress—Initial Measurement**							
**Model**	**Predictors**	R	R^2^	F	β	t	*p*
3	Low PA	0.173	0.030	8.853	−0.173	−2.975	0.003

Legend: R—simple correlation; R^2^—partial coefficient of determination; *p* (Sig.)—statistical significance; β—standardized beta coefficient; t—t-statistic; F—model significance; PA—physical activity.

**Table 5 life-15-01179-t005:** Levels of physical activity as predictors of anxiety, depression, and stress at the final measurement (n = 288).

**Anxiety—Final Measurement**							
**Model**	**Predictors**	R	R^2^	F	β	t	*p*
4	Low PA	0.139	0.019	5.611	−0.139	−2.369	0.019
	Low PA	0.193	0.037	5.492	−0.152	−2.597	0.010
	Moderate PA				−0.134	−2.300	0.022
**Depression—Final Measurement**							
**Model**	**Predictors**	R	R^2^	F	β	t	*p*
5	Total PA	0.183	0.033	9.880	−0.183	−3.143	0.002
	Total PA	0.238	0.057	8.564	−0.282	−4.109	<0.001
	Vigorous PA				0.182	2.653	0.008
**Stress—Final Measurement**							
**Model**	**Predictors**	R	R^2^	F	β	t	*p*
6	Low PA	0.134	0.018	5.226	−0.134	−2.286	0.023
	Low PA	0.179	0.032	4.736	−0.145	−2.485	0.014
	Moderate PA				−0.120	−2.047	0.042

Legend: R—simple correlation; R^2^—partial coefficient of determination; *p* (Sig.)—statistical significance; β—standardized beta coefficient; t—t-statistic; F—model significance; PA—physical activity.

**Table 6 life-15-01179-t006:** Household, work, leisure, and transport activity domains in relation to mental health at initial measurement (n = 288).

**Anxiety—Initial Measurement**							
**Model**	**Predictors**	R	R^2^	F	β	t	*p*
7	Transport	0.192	0.037	10.991	−0.192	−3.315	0.001
**Depression—Initial Measurement**							
**Model**	**Predictors**	R	R^2^	F	β	t	*p*
8	Transport	0.194	0.038	11.242	−0.194	−3.353	0.001
**Stress—Initial Measurement**							
**Model**	**Predictors**	R	R^2^	F	β	t	*p*
9	Transport	0.134	0.018	5.203	−0.134	−2.281	0.023

Legend: R—simple correlation; R^2^—partial coefficient of determination; *p* (Sig.)—statistical significance; β—standardized beta coefficient; t—t-statistic; F—model significance.

**Table 7 life-15-01179-t007:** Effects of individual physical activity domains on anxiety, depression, and stress at final measurement (n = 288).

**Anxiety—Final Measurement**							
**Model**	**Predictors**	R	R^2^	F	β	t	*p*
10	Household PA	0.164	0.027	7.861	−0.164	−2.804	0.005
	Household PA	0.220	0.048	7.253	−0.165	−2.853	0.005
	Transport				−0.147	−2.548	0.011
**Depression—Final Measurement**							
**Model**	**Predictors**	R	R^2^	F	β	t	*p*
11	Household PA	0.185	0.034	10.112	−0.185	−3.180	0.002
**Stress—Final Measurement**							
**Model**	**Predictors**	R	R^2^	F	β	t	*p*
12	Transport	0.133	0.018	5.131	−0.133	−2.265	0.024
	Transport	0.179	0.032	4.712	−0.134	−2.296	0.022
	Household PA				−0.120	−2.058	0.041

Legend: R—simple correlation; R^2^—partial coefficient of determination; *p* (Sig.)—statistical significance; β—standardized beta coefficient; t—t-statistic; F—model significance; PA—physical activity.

## Data Availability

The authors will make available the raw data that underpin the conclusions of this article upon request.

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
