# Peer review of "Physical Activity and Mental Health After COVID-19: The Role of Levels and Domains of Physical Activity"

_life, 2025, doi:10.3390/life15081179_

Round 1
Reviewer 1 Report (Previous Reviewer 1)
Comments and Suggestions for Authors
The topic has been studied a lot before. The data is too old and not representative. Only descriptive statistics presented. Need have in-depth analysis. Now is too basic analysis. Need to think about the research framework and model. Develop relevant hypotheses.
Author Response
Dear Reviewer,
Thank you very much for taking the time to review our manuscript and for your valuable contributions to its substantial improvement.
Please find the detailed responses and the corresponding revisions/corrections highlighted in the file attached.
Kind regards,
The authors

Reviewer 2 Report (Previous Reviewer 3)
Comments and Suggestions for Authors
Dear Authors,
I reviewed this manuscript in the past (another number). You suited the manuscript to my comments. Now the manuscript is methodologically correct (except for the Conclusions).
The conclusions are poorly developed. Please read the Instructions for Authors of other journals, what conclusion should contain, and what should not. Again, the conclusions may be related only to your research. This is not the place for discussion.
Conclusions: the sentence „Although high intensity PA shows potential for positively impacting depression, our findings reveal a more complex and potentially non-linear relationship that warrants further investigation”. What does „more complex” mean? The conclusions should not contain generalities but should clearly refer to the purpose of the research.
Moreover, the level of physical activity should not be confused with the level of intensity (Conclusions, the first sentence is ambiguous).
Author Response
Dear Reviewer,
Thank you very much for taking the time to review our manuscript and for contributing to its substantial improvement.
Please find the detailed responses and the corresponding revisions/corrections highlighted in the file, attached.
Kind regards,
The authors

Reviewer 3 Report (New Reviewer)
Comments and Suggestions for Authors
Please see the attachment

Author Response
Dear Reviewer,
Thank you very much for taking the time to review our manuscript and for your valuable contributions to its substantial improvement.
Please find the detailed responses and the corresponding revisions/corrections highlighted in the file attached.
Kind regards,
The authors

Round 2
Reviewer 1 Report (Previous Reviewer 1)
Comments and Suggestions for Authors
All my concerns have been addressed.
This manuscript is a resubmission of an earlier submission. The following is a list of the peer review reports and author responses from that submission.
Round 1
Reviewer 1 Report
Comments and Suggestions for Authors
There is an interesting study. No theoretical foundation. The topic has been studied a lot before. What are your unique contributions? Why these two centres data was collected? Are they represented all other areas round the world? The sample are biasd to female. When was the interviews? What was the interval? Without detail methodology information, we cannot assess the validty and reliability. Conclusion is too short? What is the managerial implication?
Author Response
Dear Reviewer,
Thank you very much for taking the time to review this manuscript and for contributing to its substantial improvement.
Please find attached the detailed responses and the corresponding revisions/corrections highlighted in the resubmitted file.
Kind regards,
The authors

Reviewer 2 Report
Comments and Suggestions for Authors
1. Study Design and Sample
The study employs a longitudinal observational design, tracking changes in physical activity (PA) and mental health over approximately three months. This design is appropriate for examining associations over time but does not permit causal conclusions. Stratified random sampling based on geographic distribution strengthens representativeness within the Jablanica District. However, the absence of a control group (e.g., individuals without prior COVID-19) limits comparative interpretation. Additionally, the lack of a priori power analysis raises questions about the study’s ability to detect effects of clinical significance. The exclusion of hospitalized individuals with severe COVID-19 further narrows generalizability.
2. Measurement Instruments
PA was assessed using the long version of the IPAQ, which, while validated, is prone to overestimation due to its self-report nature. No information is provided about cultural adaptation or psychometric validation of the IPAQ and DASS-42 in the Serbian population. Given that the DASS-42 assesses recent (past week) emotional states, its use at only two time points may not capture relevant fluctuations in mental health over time. Additionally, the study would benefit from triangulating self-report data with objective measures (e.g., wearable devices) where possible.
3. Statistical Analyses
The authors used paired t-tests to compare PA and mental health outcomes at two time points. This is an appropriate method for within-subject analysis; however, no test of normality is reported to validate the use of parametric tests. Including effect sizes (e.g., Cohen’s d) would have enhanced the understanding of clinical relevance. For predictive modeling, stepwise linear regression was used to examine the influence of PA levels and domains on DASS outcomes. While common, stepwise regression is statistically controversial due to its data-driven nature, risk of overfitting, and instability across samples. The paper does not mention whether multicollinearity diagnostics (e.g., variance inflation factor) were conducted. Moreover, the models explained only a small portion of variance in outcomes (R² values generally below 6%), and residuals diagnostics are not reported. These omissions reduce confidence in the regression findings.
4. Interpretation of Results
The authors correctly highlight that low and moderate PA were associated with better mental health outcomes, yet the findings on vigorous PA (e.g., a positive beta coefficient for depression) suggest a non-linear or U-shaped relationship. The current models are limited to linear associations. Including interaction terms or non-linear modeling (e.g., quadratic terms) could provide a deeper understanding of these patterns. Additionally, the use of change scores in regression, without adjusting for baseline values as covariates, introduces statistical artifacts and potential regression to the mean.
5. Evaluation of Results and Discussion
The results section is generally well-structured and appropriately presents the statistical outputs. However, the interpretation could benefit from a clearer distinction between statistical significance and clinical relevance. For instance, although the reduction in anxiety scores reached statistical significance (p = .047), the magnitude of change appears minimal (Δ = 0.94), raising concerns about the practical impact.
In the regression analyses, although some PA domains and intensities emerge as significant predictors, the low R² values across all models (ranging between 1.8% and 5.7%) indicate that most of the variance in mental health outcomes remains unexplained. This limitation is not sufficiently emphasized in the discussion.
The discussion section makes an effort to contextualize findings within the literature, especially regarding the differential impact of PA intensity and domain. The authors rightly note the potential U-shaped relationship between vigorous PA and depression, but this hypothesis is not statistically tested in the current models. The suggestion that household and transport-related PA have distinct psychological benefits is valuable, yet further elaboration on possible mechanisms (e.g., routine, autonomy, purpose) would enrich the interpretation.
Lastly, while limitations are acknowledged, the discussion could be more critical regarding methodological issues—such as self-report biases, the risk of Type I error due to multiple comparisons, and the use of stepwise regression without cross-validation or robustness checks. These are key concerns that should temper the strength of the conclusions drawn.
6. Conclusion
This study addresses an important public health issue and is commendable for its dual-time-point design and domain-specific PA analysis. However, to strengthen the validity and applicability of the findings, greater methodological rigor and transparency in statistical reporting are necessary.
Author Response

(The authors gave the same response as above.)

Reviewer 3 Report
Comments and Suggestions for Authors
Dear Authors,
you conducted correct methodological research. Detailed comments:
1. Introduction – the justification for conducting the research was poorly presented.
2. The number of people surveyed is not large, as for survey research. Were calculations made for the “minimum sample”? 3. The research was conducted 3 years ago. Why was it sent for publication only in 2025?
4. Methods: survey research, to be credible, should be anonymous. The research was conducted via a telephone interview. Please provide justification for this form of research for your research.
5. Table 2. References were not provided. These are not your calculation formulas. 6. Conclusions: developed incorrectly. Please refer them in detail to your research objective. Line 457-461 are not conclusions, but strengths and limitations or discussion.
Author Response

(The authors gave the same response as above.)
